# UNCLIPPING CLIP'S WINGS: AVOIDING ROBUSTNESS PITFALLS IN MULTIMODAL IMAGE CLASSIFICATION

## ABSTRACT

Despite being pretrained on large-scale data, multimodal models such as CLIP can still learn spurious correlations. However, CLIP does not seem to learn the same spurious correlations as standard vision models, performing worse on some benchmark datasets (Waterbirds) yet better on others (CelebA). We investigate this discrepancy and find that CLIP's robustness on these datasets is highly sensitive to the choice of class prompts. Worst-group accuracy can be arbitrarily improved or worsened by making minute, single-word changes to prompts. We further provide evidence that the root cause of this phenomenon is *coverage* — using class prompts that are out-of-distribution with respect to pretraining can worsen spurious correlations. Motivated by these findings, we propose using class prompts that are generated from a public image-to-text model, such as BLIP. We show that performing $k$-nearest neighbors on these prompt embeddings improve downstream robustness without needing to fine-tune CLIP.

## 1 INTRODUCTION

Icarus, who perished by flying too close to the sun, made the fatal mistake of ignoring distribution shift — namely, that proximity to the sun would increase ambient temperature, melting the wax that held his wings together.[1] Much like Icarus' wings, we too desire our machine learning models today to be *robust* — an umbrella term that describes the model's ability to maintain good performance in the face of distribution shifts at test time.

Of the many flavors of distribution shifts that have been studied, one such pernicious phenomenon is the presence of *spurious correlations* or shortcuts. These are features that are highly correlated to the label under the training distribution, however, this relationship breaks down on unseen test distributions. One canonical example is the image background, with distribution shift occuring when objects are photographed at a different place or time of day (Beery et al., 2018; Zech et al., 2018). Models trained under empirical risk minimization (ERM) have been observed to rely on a combination of shortcut and salient features and therefore fail to generalize at test time (Hovy & Søgaard, 2015; Hashimoto et al., 2018; Puli et al., 2023).

In recent years, large pretrained models have enjoyed notable success on a wide range of tasks. These refer to models with billions of parameters that are typically trained in a self-supervised manner on broad Internet-scale data (Bommasani et al., 2021). The resulting model can then be adapted to downstream applications by fine-tuning the model's parameters on the smaller dataset of interest. For image classification tasks, the most widely-used pretrained model is CLIP (Radford et al., 2021), whereby an image and text encoder are jointly trained using the contrastive InfoNCE objective (Oord et al., 2018) on a huge corpus of image-caption pairs. Downstream classification is performed in a zero-shot manner by codifying class labels as text prompts, then predicting the class whose label embedding has the highest inner product with the image embedding.

It was hoped that CLIP and other large pretrained models would be more robust to spurious correlations than smaller bespoke vision models, having been exposed to data orders of magnitude larger than the downstream dataset containing the spurious correlation. Results, however, paint a more

---

[1]We know today that temperature actually decreases as altitude increases. Of course, one should never let scientific inaccuracy get in the way of good storytelling.

complicated picture. Zhang & Ré (2022) show that the gap between average accuracy and worst-group accuracy ranged from 55.6% in Waterbirds (Wah et al., 2011; Sagawa et al., 2019) to 7.9% in CelebA (Liu et al., 2018), two widely-used benchmark datasets. Clearly, large-scale pretraining is not the one-stop solution for mitigating spurious correlations.

Existing literature has simply sought to remedy this gap by fine-tuning CLIP on the biased training dataset (Zhang & Ré, 2022; Yang et al., 2023), either through explicitly using additional labels of the spurious feature or by making certain assumptions of model behavior. These methods can indeed be viewed as contrastive analogues of methods originally proposed in the targeted setting (Sagawa et al., 2019; Liu et al., 2021).

Motivated instead by the unexplained difference in worst-group accuracy between CelebA and Waterbirds, we set out to more closely probe zero-shot behavior in these datasets. We expand our experiments to include OpenCLIP (Ilharco et al., 2021) in addition to the original CLIP models. We observe that CLIP's zero-shot prediction of *background* in Waterbirds is just as poor as foreground. Further probing of image embedding space shows poor separability by both foreground and background features. These observations cannot be fully explained by spurious correlations.

Our **first contribution** is to show that the choice of class prompt greatly affects zero-shot accuracy in both CelebA and Waterbirds. Arbitrary changes to prompt templates can worsen or improve worst-group performance. Delving deeper, we show that the root cause of this discrepancy is due to the class prompts being *out-of-distribution* (OOD) during CLIP's pretraining. We verify this directly on OpenCLIP and MetaCLIP (Xu et al., 2023) by counting token frequencies.

From these experiments, we conclude that choosing in-distribution class prompts that CLIP has seen during pretraining is critical to zero-shot success, particularly in datasets containing spurious correlations where OOD prompts can reinforce such biases. To this end, our **second contribution** is leveraging the use of large, public image-to-text models to automatically generate proxy class prompts for downstream classification.

We show that such a model — we use BLIP (Li et al., 2022a) in our experiments — can be used to generate captions on the downstream dataset, which are then used to classify test samples via $k$-nearest neighbors. Our approach achieves comparable results on spurious correlation datasets *without* needing to fine-tune CLIP's embeddings on the downstream dataset and *without* requiring any spurious labels. We verify our method on ImageNet-1K (Deng et al., 2009) in addition to Waterbirds and CelebA. Beyond robustness, our work is also a step towards automating downstream classification without requiring human input to generate class prompts.

## 2 BACKGROUND AND PROBLEM SETUP

For a downstream image classification task, we let $\mathbf{x} \in \mathcal{X}$ denote covariates, $y \in \mathcal{Y}$ the class label and $s \in \mathcal{S}$ the spurious label. We consider a family of data-generating distributions $p_e(\mathbf{x}, y, s)$ indexed by the environment $e$, of which the training ($e = tr$) and test ($e = te$) distributions are two such environments. Spurious correlations happen when $p_e(y, s)$ changes across environments.

Most spurious correlation datasets contain salient features $\mathbf{h} := \mathbf{h}(\mathbf{x})$ that can predict $y$ perfectly. That is, there exist some deterministic function $f_1$ such that $f_1(\mathbf{h}) = y$ for all $e$. Furthermore, there is no deterministic function $f_2$ such that $f_2(\mathbf{h}) = s$ for all $e$. The existence of such $\mathbf{h}$ implies that $p_{tr}(y|\mathbf{h}) = p_{te}(y|\mathbf{h})$. As such, the optimal predictor that minimizes training loss will also minimize test loss and the Bayes optimal predictor $p_{tr}(y|\mathbf{x})$ should be robust to test-time distribution shift. Unfortunately, empirical risk minimization (ERM) generally fails to learn $\mathbf{h}$, instead learning a representation of $s$ that breaks at test time (Sagawa et al., 2019; Geirhos et al., 2020).

As $y$ and $s$ typically have discrete support, we denote their Cartesian product $g = (y, s)$ as the *group*. Colloquially, we use the terms "majority group" and "minority group" to refer to groups with disproportionate representation in the training distribution. In addition to average accuracy $\mathcal{A}_{ave}$ on the test distribution, we also evaluate *worst-group accuracy* (WGA) across all groups:

$$\mathcal{A}_{worst}\big(p_{te}(\mathbf{x}, y)\big) = \min_g \mathcal{A}_{ave}\big(p_{te}(\mathbf{x}, y|g)\big) \tag{1}$$

**CLIP** CLIP consists of an image encoder $f_\theta$ and a text encoder $g_\varphi$, trained jointly with respect to a pretraining distribution $q_{pt}(\mathbf{x}, \mathbf{t})$ of image-caption pairs $(\mathbf{x}, \mathbf{t})$. CLIP is trained in a contrastive

| Method | Waterbirds | | | CelebA | | |
|---|---|---|---|---|---|---|
| | WG | Average | Gap | WG | Average | Gap |
| ERM ResNet-50 (Sagawa et al., 2019) | 60.0 | 97.3 | 37.3 | 41.1 | 94.8 | 53.7 |
| CLIP ResNet-50 | 39.3 | 77.2 | 38.0 | 82.2 | 87.9 | 5.7 |
| CLIP ViT-L/14 | 45.2 | 84.4 | 39.2 | 74.3 | 80.7 | 6.5 |
| OpenCLIP ViT-L/14 | 46.3 | 73.7 | 27.5 | 15.6 | 89.0 | 73.5 |
| CLIP ResNet-50 Spurious Prediction | 52.8 | 71.9 | 19.1 | 89.4 | 98.9 | 9.4 |
| CLIP ViT-L/14 Spurious Prediction | 55.7 | 75.1 | 19.4 | 92.8 | 99.0 | 6.3 |
| OpenCLIP ViT-L/14 Spurious Prediction | 72.3 | 83.5 | 11.2 | 90.0 | 99.0 | 9.0 |

Table 1: Worst-group and average zero-shot accuracies on Waterbirds and CelebA test sets. In rows 2-4 we predict the true label; in rows 5-7 we predict the spurious attribute. For comparison, row 1 shows the vanilla ERM results on a single ResNet-50 network, taken from Sagawa et al. (2019).

manner using the InfoNCE objective. For a given minibatch $\{\mathbf{x}_i, \mathbf{t}_i\}_{i=1}^N$ of size $N$, we have:

$$\mathcal{L}_{CLIP}(\theta, \varphi) = -\frac{1}{2}\mathbb{E}_{i \sim \mathcal{U}[1,...,N]}\Big[\frac{e^{\langle f_\theta(\mathbf{x}_i), g_\theta(\mathbf{t}_i)\rangle/\tau}}{\sum_{j=1}^N e^{\langle f_\theta(\mathbf{x}_i), g_\theta(\mathbf{t}_j)\rangle/\tau}}\Big] \tag{2}$$

$$-\frac{1}{2}\mathbb{E}_{i \sim \mathcal{U}[1,...,N]}\Big[\frac{e^{\langle f_\theta(\mathbf{x}_i), g_\theta(\mathbf{t}_i)\rangle/\tau}}{\sum_{j=1}^N e^{\langle f_\theta(\mathbf{x}_j), g_\theta(\mathbf{t}_i)\rangle/\tau}}\Big] \tag{3}$$

where $\tau$ is a temperature hyperparameter. Once trained, zero-shot downstream classification can be done: For a given image dataset $\{\mathbf{x}, y\}_{i=1}^N$, we encode all images into embeddings $f_\theta(\mathbf{x})$. We also encode class labels as text embeddings by first manually describing the classes, and then filling this description into commonly-used *class prompt templates*. An example of a class prompt template is *"This is the photo of a [class_name]."*. This prompt, which we denote as $\mathbf{t}_y$, is then encoded into embeddings $g_\varphi(\mathbf{t}_y)$. To reduce notational clutter, thereafter we will use $\mathbf{x}$ and $\mathbf{t}$ to refer to embeddings $f_\theta(\mathbf{x})$ and $g_\varphi(\mathbf{t})$ respectively where unambiguous. For an image $\mathbf{x}$, we predict the class with the largest inner product of embeddings:

$$\hat{y} = \arg\max_{c \in \mathcal{Y}} \langle \mathbf{x}, \mathbf{t}_c \rangle \tag{4}$$

**Datasets** Waterbirds (Sagawa et al., 2019) and CelebA (Liu et al., 2018) are two benchmark datasets for spurious correlations. Waterbirds is made by artificially superimposing 200 species of birds (terrestrial and aquatic) from the Caltech-UCSD Birds-200-2011 dataset (Wah et al., 2011) on four backgrounds from the Places dataset (Zhou et al., 2017). The binary classes are $\mathcal{Y} = \{\text{landbird}, \text{waterbird}\}$ and the spurious correlation is the background $\mathcal{S} = \{\text{land background}, \text{water background}\}$. The training dataset largely contains images of birds in their natural habitats, hence the minority groups are landbirds on water and vice versa. CelebA is a natural image dataset of celebrity faces. The class attribute is hair color $\mathcal{Y} = \{\text{blond}, \text{not blond}\}$ and the spurious attribute is gender $\mathcal{S} = \{\text{male}, \text{female}\}$. The minority group is blond men.

## 3 RELATED WORK

**CLIP and its variants** Contrastive Language-Image Pretraining (CLIP) (Radford et al., 2021) pioneered the use of contrastively matching (Oord et al., 2018) image-caption pairs as an effective, at-scale pretraining task to learn useful image representations for downstream tasks. Later works have extensively studied CLIP's effectiveness and proposed various improvements. Some have tangential relevance to robustness, for example, a fine-grained variant that matches regions of the image to specific textual concepts (Zhong et al., 2022), exploiting Hopfield networks to encourage the encoder to extract richer features from the image (Fürst et al., 2022), and performing max-pooling in CLIP's vision encoder to reduce background bias (Li et al., 2022b). Petryk et al. (2022) uses CLIP to improve the robustness of a vision model by guiding it to use specific parts of the image.

**Spurious Correlations and Shortcut Learning** Distribution shifts in the form of spurious correlations that do not hold at test-time were identified by Beery et al. (2018), Zech et al. (2018), and

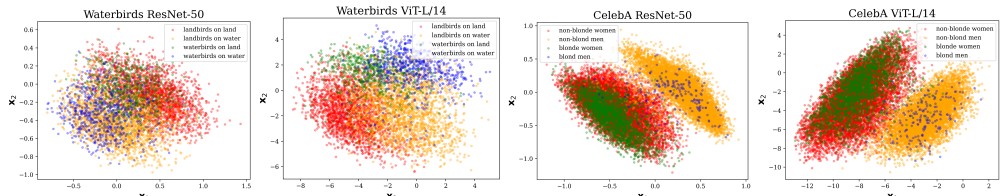

Figure 1: 2D projections of CLIP image embeddings for the Waterbirds and CelebA test sets, colored by group. For both datasets, the salient feature is not a principal component. However, for CelebA only, the spurious feature (gender) is a principal component. There are no major differences between embedding projections produced by the two architectures. Figure 3 in Appendix B shows the OpenCLIP embeddings, which follow the same trends as the CLIP models here.

Buolamwini & Gebru (2018), amongst others, and more comprehensively formalized and studied later on by works such as Geirhos et al. (2020) and Moayeri et al. (2022). Literature can broadly be divided into two categories. If salient features $h(x)$ do not exist, we must rely on additional assumptions, such as counterfactual invariance (Veitch et al., 2021), access to unlabelled data from the test distribution (Sun et al., 2022), or access to training group labels to learn shortcut-independent representations (Puli et al., 2021). In the setting (which we consider) where $h(x)$ exists, methods to learn $h$ assume access to training group labels (Sagawa et al., 2019), last-layer fine-tuning (Kirichenko et al., 2022), treating initially misclassified training examples as a proxy for minority groups (Liu et al., 2021), or exploiting the information bottleneck in generative models (Yang et al., 2022). More recently, works have investigated *why* ERM fails to learn $h$, proposing margin-related inductive biases as the root cause (Puli et al., 2023).

**Spurious Correlations in CLIP**   Zhang et al. (2022) first proposed a contrastive approach for mitigating spurious correlations in (unimodal) vision models, whereby spurious attributes are inferred through Liu et al. (2021) and then used to learn similar representations between majority and minority samples from the same class. Zhang & Ré (2022) follow up by identifying spurious correlations as a problem in CLIP specifically, and propose fine-tuning CLIP embeddings in such a contrastive manner. Yang et al. (2023) also propose fine-tuning CLIP embeddings contrastively but they make explicit use of group labels. Unlike these earlier work, our paper is the first to (i) highlight that CLIP's poor performance is due in part to OOD class prompts, and (ii) propose a method of improving zero-shot classification *without* fine-tuning embeddings on downstream datasets.

Finally, recent work by Adila et al. (2023) also aim to improve CLIP's robustness without fine-tuning. They query a large language model (LLM) for additional knowledge of the salient feature to adjust CLIP's embeddings. Our work is similar to theirs as we also propose augmenting CLIP with a publicly-available large pretrained model — namely, the image-to-text model BLIP (Li et al., 2022a). However, our proposed system is much more automated as their approach requires extensive manual input (knowing the right queries to ask the LLM).

## 4 SPURIOUS CORRELATIONS IN CLIP: AN INVESTIGATION

To better understand how and why spurious correlations are learnt by CLIP, we began by examining existing results on Waterbirds and CelebA. Table 1 shows worst-group (WGA) and average accuracies on Waterbirds and CelebA for three models: (i) CLIP with the ResNet-50 image encoder, (ii) CLIP with the ViT-L/14 image encoder, and (iii) OpenCLIP with the ViT-L/14 image encoder. For (i) and (ii), we use the official implementation by (Radford et al., 2021). For (iii), we use the model trained with the LAION-400M dataset, the same size as CLIP's pretraining corpus. OpenCLIP does not have a ResNet-50 encoder.

CelebA WGA is significantly lower than Waterbirds WGA on a standard ERM model. However, this trend reverses completely on both CLIP architectures, with the former *exceeding* the latter by roughly 30% to 40%. Furthermore, the worst-group gap (the difference between WGA and average accuracy) on CelebA is only ~5%, suggesting that the model has not learnt spurious correlations at all. Even more bewildering, this result cannot be replicated for the OpenCLIP ViT-L/14 model. On

| Waterbirds | CLIP ResNet-50 | | | CLIP ViT-L/14 | | | OpenCLIP ViT-L/14 | | |
|---|---|---|---|---|---|---|---|---|---|
| | WG | Average | Gap | WG | Average | Gap | WG | Average | Gap |
| True Label Prediction (from Table 1) | 39.3 | 77.2 | 38.0 | 45.2 | 84.4 | 39.2 | 46.3 | 73.7 | 27.5 |
| Species Prediction | 18.5 | 24.6 | 6.1 | 29.4 | 35.0 | 5.7 | 43.0 | 46.1 | 3.1 |
| Species Prediction (Top-5) | 43.6 | 53.2 | 9.6 | 62.6 | 69.2 | 6.6 | 69.7 | 74.7 | 5.0 |
| Species Binarized | **72.6** | **86.9** | 14.3 | **82.6** | **94.8** | 12.3 | **82.2** | **92.0** | 9.7 |
| Background Prediction (from Table 1) | 52.8 | 71.9 | 19.1 | 55.7 | 75.1 | 19.4 | **72.3** | 83.5 | 11.2 |
| Location Prediction | 46.0 | 63.7 | 17.7 | 60.1 | 74.7 | 14.6 | 60.9 | 80.0 | 19.2 |
| Location Binarized | **69.2** | 92.9 | 23.7 | **74.6** | 91.9 | 17.3 | **72.4** | **92.8** | 20.4 |

Table 2: Results of zero-shot classification on Waterbirds for fine-grained foreground (species) and background (location) attributes. **Rows 1 and 5:** Accuracy on the original binary label and spurious attribute taken from Table 1, shown here for comparison. **Rows 2 and 6:** Accuracy on the fine-grained attributes. **Row 3:** For species, we also report the percentage of samples where the correct class is one of the top 5 inner products (out of 200 classes). **Rows 4 and 7:** Accuracy where the predicted fine-grained attribute is mapped back to the original binary categories.

the OpenCLIP implementation, the CelebA WGA is a paltry 15%, even lower than standard ERM. The only difference between the CLIP and OpenCLIP implementations is the pretraining dataset.

These results are seemingly inexplicable when we consider the standard narrative of spurious correlations. Recall that the predictive equivalent of the InfoNCE objective is the cross-entropy loss. Spurious correlations learnt by an ERM model trained via cross-entropy loss, as is the case in Sagawa et al. (2019), will also be learnt by contrastive models like CLIP. As such, the use of contrastive learning alone does not explain why CelebA WGA improved so drastically.

The obvious and immediate suspect is pretraining support. Table 1 suggests that both CLIP and OpenCLIP's pretraining datasets are biased towards majority samples in the case of Waterbirds, resulting in consistent worst-group gaps across all three models. Conversely, for CelebA, we might reason that OpenCLIP's dataset is strongly biased whereas CLIP's dataset contains a sizeable number of majority and minority samples alike, explaining the discrepancy in WGA between CLIP and OpenCLIP. However, as we will show in further experiments, this explanation too is inadequate.

### 4.1 SPURIOUS ATTRIBUTE PREDICTION

It is not possible to directly compare the two pretraining datasets without access to CLIP's pretraining dataset. Instead, our first proxy is to establish how strongly each model has learnt the spurious concept. We perform zero-shot classification using the *spurious attribute as label*, i.e. predicting background on Waterbirds and gender on CelebA. Table 1 (last three rows) shows these results.

Our findings are counterintuitive. On the two CLIP models, the Waterbirds WGA is ∼50% — no better than random and only slightly higher than true label prediction. In other words, **CLIP's zero-shot performance on background prediction is almost as poor as foreground (label) prediction**. If CLIP's pretraining distribution was skewed towards majority groups and had allowed the model to learn background as a spurious correlation, we would accordingly expect the encoder to learn a strong representation of background features. However, our results show that CLIP is *unable* to (correctly and non-spuriously) associate background features with their label.

Furthermore, we observe that the *average* accuracy has also decreased by 5% to 10% compared to true label prediction. This indicates that background prediction is relatively poorer for majority groups than minority groups — further contradicting the naive explanation that pretraining coverage of Waterbirds is biased towards majority groups.

### 4.2 EXAMINING THE IMAGE EMBEDDING SPACE

To further support this point, we visually examine CLIP's image embeddings on both datasets. Figure 1 plots the first two principal components of images embeddings, split by group. In Waterbirds, neither foreground nor background correspond to principal directions of separability. As such, **poor**

| CelebA | CLIP ResNet-50 | | | CLIP ViT-L/14 | | | OpenCLIP ViT-L/14 | | |
|---|---|---|---|---|---|---|---|---|---|
| | WG | Average | Gap | WG | Average | Gap | WG | Average | Gap |
| Original: . . . *celebrity with { blond, no blond } hair.* | **82.2** | 87.9 | 5.7 | 74.3 | 80.7 | 6.5 | 15.6 | 89.0 | 73.5 |
| . . . *celebrity with { blond, non-blond } hair.* | 54.8 | 88.9 | 38.8 | 26.3 | 87.2 | 60.9 | 15.6 | 88.9 | 73.3 |
| . . . *celebrity whose hair is { blond, not blond }.* | 58.7 | 82.8 | 24.2 | **80.4** | 85.4 | 5.0 | **58.9** | 90.9 | 32.0 |
| . . . *human with { blond, no blond } hair.* | **82.2** | 87.9 | 5.7 | 59.3 | 71.1 | 11.7 | 53.5 | 63.1 | 9.6 |

Table 3: Zero-shot classification on CelebA with various class prompts. Even minute differences in the prompts, e.g. changing *"not blond"* to *"non-blond"* result in significant drops of WGA. There is also little correlation in WGA between the three models.

**foreground prediction cannot be fully explained by the model having learnt background as a spurious correlation**, corroborating our findings in Sections 4.1.

Conversely, CelebA images are well-separated by the spurious attribute but not the salient feature. The fact that (i) the two CLIP models perform well (∼80% WGA) on CelebA, and yet (ii) do not produce image embeddings that are separable by class is our first clue that the choice of text prompt plays a significant role. A further indication that the naive spurious correlation explanation does not hold comes from the OpenCLIP image embeddings *(plots shown in Appendix B Figure 3 due to space constraints)*. Despite the vast difference in WGA between the CLIP and OpenCLIP ViT-L/14 models, both models produce almost identical image embeddings — separable by the spurious feature but not the true label. It is clear that we must examine the *text* component if we are to explain these findings adequately.

4.3 VARYING CLASS PROMPTS IN ZERO-SHOT CLASSIFICATION

We performed a series of experiments where we varied the class prompts used at test-time. In both datasets, we found that changing the class prompts significantly affected zero-shot accuracy.

**Waterbirds**  As noted in Section 2, Waterbirds was made by artificially superimposing natural images of 200 species of birds on four types of backgrounds (`bamboo forest`, `forest`, `lake`, `ocean`). These fine-grained attributes, which we denote as "species" and "location", were binarized into { `land`, `water` } to form the final dataset. We consider two sets of experiments: (1) direct zero-shot classification using species and location attributes as labels, and (2) we take the results of (1) and manually map the predicted species or location to their binary category { `landbird`, `waterbird` } or { `land background`, `water background` } respectively.

Table 2 shows the results of these experiments. For the foreground, species prediction (a $K = 200$ classification problem) is reasonably worse than label prediction, however, by simply expanding the margin of error to the top five classes, we find that all three models already outperform true label prediction. In other words, CLIP (and OpenCLIP) has a higher rate of success narrowing down the bird species to five of 200 possible choices than it has classifying the image as landbird or waterbird. The same behavior is true for background — CLIP is better at predicting one of four exact locations than the binarized land or water background.

Furthermore, when we take the fine-grained attribute that the models predict and manually map it back to the original binary categories, we find that CLIP's performance improves even further, with WGA improving up to 82% on the two ViT-L/14 architectures. This implies that CLIP has a much more robust understanding of the foreground feature than its WGA suggests, yet struggles with the simpler task of predicting one of two broad categories.

**CelebA**  We design several variants of the original class prompts used by Zhang & Ré (2022) (as reported in Table 1). Table 3 shows the results of zero-shot classification on each set of prompts. We see that minute differences in the choice of class prompt lead to drastic drops in WGA. For example, simply changing the phrase from *"no blond hair"* to *"non-blond hair"* reduces WGA by 30-50% on CLIP. We also see little correlation in the results of the three models: each model performs best on a different prompt, and what improves WGA on one model can worsen WGA on another.

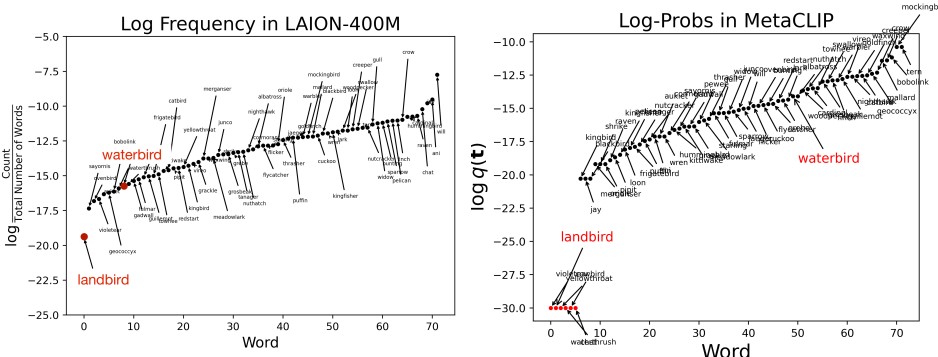

Figure 2: **(a)** Log frequencies of the various bird species on LAION-400M, the pretraining dataset for OpenCLIP. Both *"waterbird"* and *"landbird"* have significantly lower counts than the vast majority of bird species. Specifically, *"landbird"* occurs the least frequently. **(b)** Marginal log probabilities for the same prompts on MetaCLIP. Points in red denote words with zero probability in the pretraining distribution. We see that *"waterbird"* has lower probability than about half of the bird species and *"landbird"* has zero probability.

In both datasets, we see that **zero-shot accuracy is highly sensitive to the choice of prompt**. CelebA WGA can be arbitrarily worsened by making minute, semantic-preserving changes to class prompts. Conversely, Waterbirds WGA improved on the *harder* task of predicting more fine-grained attributes. These findings undermine the conventional understanding that spurious correlations are the sole reason for poor performance on minority groups.

### 4.4 OUT-OF-DISTRIBUTION DETECTION OF CLASS PROMPTS

Spurious correlations alone fail to explain our findings above. How, then, can we resolve the discrepancy between the expected and observed WGAs in these benchmark datasets? Our hypothesis is that choosing **class prompts that are OOD with respect to CLIP's pretraining distribution** significantly impairs zero-shot accuracy. In datasets where spurious correlations occur, this effect can arbitrarily reinforce or mitigate the worst-group gap.

To verify such a claim, we need to compute the marginal likelihood $q_{pt}(\mathbf{t})$ of the text prompts used in these datasets. This is not directly possible as the data that CLIP was pretrained on is not publicly available. They cannot be estimated from the inner products $\langle \mathbf{x}, \mathbf{t} \rangle$ of image-text embeddings either, as these products are *ratios* of joint and product-of-marginal distributions $\frac{q_{pt}(\mathbf{x}, \mathbf{t})}{q_{pt}(\mathbf{x})q_{pt}(\mathbf{t})}$ and so individual marginal distributions cannot be extracted from them. Other methods that try to estimate these likelihoods can also present pitfalls (Zhang et al., 2021). Instead, we rely on open-source versions of CLIP to perform this analysis.

**OpenCLIP on Waterbirds**  On OpenCLIP, we can directly count the frequencies of class tokens on LAION-400M, its pretraining dataset. Figure 2a compares the log-frequency of the tokens representing each of the individual bird species, as well as the words *"landbird"* and *"waterbird"*. We see that the tokens for *"waterbird"* and *"landbird"* have lower frequencies than the tokens representing the vast majority of bird species. In particular, we see that the *"landbird"* token is exceedingly rare, being about *three magnitudes less frequent than the next rarest token*.

**MetaCLIP on Waterbirds**  We also consider MetaCLIP Xu et al. (2023), an effort to mimic the pretraining distribution of CLIP by balancing an open-source dataset (400M Common Crawl image-text pairs) on CLIP's metadata, and subsequently training on the balanced dataset. We analyze the datacard of the `ViT-L-14-quickgelu` model and plot the log marginal likelihoods of the same terms in Figure 2b. We see that *"waterbird"* has lower probability than about half of the bird species. Furthermore, the word *"landbird"* has *zero* probability, i.e. it is not in the pretraining support at all.

---

**Algorithm 1** BLIP-CLIP Image Classification

---

**Input:** training dataset $\mathcal{D}_{tr} = \{\mathbf{x}_i, y_i\}_{i=1}^{N_{tr}}$, test dataset $\mathcal{D}_{te} = \{\mathbf{x}_j\}_{j=1}^{N_{te}}$, CLIP encoders $(f, g)$,
BLIP model $b$, BLIP preamble $\mathbf{t}_1$, hyperparameter $k$
**for** $i = 1$ **to** $N_{tr}$ **do**
    $\mathbf{w}_i := g\big(b(\mathbf{x}_i|\mathbf{t}_1)\big)$
**end for**
**for** $j = 1$ **to** $N_{te}$ **do**
    {Variant 1: Image-to-Caption $k$-NN}
    $\{i_{[1]}, \ldots, i_{[k]}\} = \arg_k \max_{i \in \{1,\ldots,N_{tr}\}} \langle f(\mathbf{x}_j), \mathbf{w}_i \rangle$
    {Variant 2: Caption-to-Caption $k$-NN}
    $\{i_{[1]}, \ldots, i_{[k]}\} = \arg_k \max_{i \in \{1,\ldots,N_{tr}\}} \langle g\big(b(\mathbf{x}_j|\mathbf{t}_1)\big)\mathbf{w}_i \rangle$
    predict $\hat{y}_j \leftarrow \mathbb{I}\big[\text{ave}\{y_{i_{[1]}}, \ldots, y_{i_{[k]}}\} \geq 0.5\big]$
**end for**

---

Both OpenCLIP's frequencies and MetaCLIP's likelihoods corroborate each other and suggest that *"waterbird"* and *"landbird"*, the widely-used class prompts for the Waterbirds dataset, **are OOD with respect to pretraining**. This presents a possible explanation of our earlier findings in Section 4.3. The use of OOD class prompts lead to undefined predictive behavior. In a spurious dataset like Waterbirds, they have *exacerbated* the direction of spuriousness, resulting in even lower WGA than naive ERM. Conversely, the models perform better when we use the fine-grained species as class prompts, as these tokens are represented in pretraining. The degree of spuriousness learnt by the model is far lower than zero-shot results with OOD prompts imply.

## 5   AUTOMATING CLASS PROMPTS USING IMAGE-TO-TEXT GENERATION

Our findings in Section 4 highlight the necessity of using class prompts that have pretraining support. This begets the key question: how do we ensure that the prompts we use for downstream classification are in-distribution? Our proposal is simple. Much like we used Llama-2 as a proxy to to approximate the marginal distribution of specific class prompts, we can similarly leverage a large pretrained model to generate class prompts, under the same assumption that pretraining on large-scale data would ensure similar support over joint image-text space.

Instead of manually generating $K$ prompts (one for each class), we propose using a separate image-to-text model to generate $N_{tr}$ captions, one for each sample of the downstream training set. These captions are passed through CLIP's text encoder to be converted into text embeddings, resulting in a set of $N_{tr}$ embeddings: $\{\mathbf{t}_{\mathbf{x}_i}\}_{i=1}^{N_{tr}}$. For a given test image $\mathbf{x}^*$, prediction is carried out by performing $k$-nearest neighbors ($k$-NN) algorithm on $\mathbf{t}_{\mathbf{x}^*}$ and the support set $\{\mathbf{t}_{\mathbf{x}_i}\}_{i=1}^{N_{tr}}$. We experiment with two variants of this approach: (1) performing $k$-NN on the image embedding of $\mathbf{x}^*$, i.e. by passing the test image into CLIP's image encoder, and (2) performing $k$-NN on the *text* embedding of $\mathbf{x}^*$, i.e. by passing test image into the image-to-text model, and then converting the resulting caption into a text embedding via CLIP's text encoder. The full algorithm (both variants) is shown in Algorithm 1. We use BLIP (Li et al., 2022a), a widely-used and publicly available captioning model, for our experiments. We informally dub this approach as **BLIP-CLIP**.

We verify the performance of BLIP-CLIP in several experiments below.

1. We test on **Waterbirds** to confirm that using BLIP-CLIP circumvents the OOD text prompt issue described in Section 4 and mitigates the harmful spurious correlations. We verify that WGA has improved.

2. Our findings in Section 4 suggest that BLIP-CLIP can be useful even in datasets without spurious correlations, so long as the dataset contains a distribution shift *due to OOD text prompts*. We design such an experiment on **ImageNet-1K**.

3. We perform some ablations. We ablate on **CelebA**, where OpenCLIP results are poor (even though CLIP's baselines are excellent), and show that BLIP-CLIP improves WGA. We also ablate for using different templates during the zero-shot evaluation process, which is a common procedure when classifying with CLIP models.

| | ImageNet-1K | | | Waterbirds | | |
|---|---|---|---|---|---|---|
| | Worst-Class | Average | Gap | Worst-Group | Average | Gap |
| Baseline Zero-Shot | 42.8 | 64.6 | 21.8 | 39.3 | 77.2 | 38.0 |
| Contrastive Adapter (Zhang & Ré, 2022) | - | - | - | 86.9 | 96.2 | 9.3 |
| Yang et al. (2023): $\mathcal{L}_{lc} + \mathcal{L}_{vc} + \mathcal{L}_{vs}$ | 70.4 | 75.4 | 5.1 | **90.5** | 96.9 | 6.4 |
| BLIP-CLIP Image-to-Caption $k$-NN | **83.9** | 89.2 | 5.3 | 60.7 | 86.2 | 25.5 |
| BLIP-CLIP Caption-to-Caption $k$-NN | 77.4 | 82.1 | 4.7 | 70.7 | 80.4 | 9.7 |

Table 4: Results of BLIP-CLIP, along with existing methods for comparison. Note that even though BLIP-CLIP does not surpass the fine-tuned methods on Waterbirds, it is still able to improve upon vanilla zero-shot classification by $\sim$20% WGA *without needing any fine-tuning or spurious labels*.

## 5.1 Experimental Details And Results

**Spurious Correlations: Waterbirds**   We test our approach on the Waterbirds dataset and report our results on Table 4. We show results of both variants of BLIP-CLIP, detailed above and in Algorithm 1. We note that it is still necessary to pass a preamble prompt into BLIP. For this experiments, the preamble prompt that we pass into BLIP for completion is *"This is a picture of the bird called a"*. In addition to vanilla zero-shot classification, we also report the existing results of Zhang & Ré (2022) and Yang et al. (2023). As noted in Section 3, both of these methods fine-tune CLIP embeddings on the training dataset. They also make use of spurious attribute labels — Zhang & Ré (2022) requires spurious annotations on the validation set and Yang et al. (2023) requires spurious annotations on the test set.

From Table 4, we see that BLIP-CLIP does not surpass fine-tuned methods in WGA. However, it is still able to **bring a $\sim$20% improvement in WGA compared to vanilla zero-shot classification**. This improvement comes solely from the use of BLIP as a prompt-generating model. In particular, we stress that unlike the other methods, BLIP-CLIP **does not require fine-tuning or spurious attribute labels**.

**OOD Text Prompts: ImageNet-1K**   Our findings in Section 4 suggest that BLIP-CLIP can be extended beyond spurious correlations to OOD tasks more generally, so long as the distribution shift is due to text prompts. To verify this intuition, and to validate our approach on a natural image dataset, we also present an experiment on the ImageNet-1K dataset. We design an experiment as such: We consider 13 of the 1000 classes in the dataset that correspond to cats. All cats (family Felidae) are split into two subfamilies. 5 of these 13 families are of the subfamily Pantherinae (the "big cats"): *leopard*, *snow leopard*, *jaguar*, *lion*, *tiger*. The remaining 8 are of the subfamily Felinae: *tabby*, *tiger cat*, *Persian cat*, *Siamese cat*, *Egyptian cat*, *cougar*, *lynx*, *cheetah*. We consider a binary classification task corresponding to these two labels. We present zero-shot results as well as our method (BLIP-CLIP). Since the validation set only contains 50 samples of each class, we use the training set here for evaluation (6500 samples in the first class, and 10400 samples in the second).

We choose this setup specifically because (similar to the Waterbirds dataset) it is difficult to design in-distribution class prompts for this task, as we might expect technical taxonomic terms such as Pantherinae or Felinae to be OOD. To prove this, we experiment with a variety of prompts that a human practitioner might conceivably think of, including using the actual scientific terms and using layman terms (big cat vs small cat). The baseline results in Table 4 show the best WGA amongst the various such prompts. BLIP-CLIP surpasses all baselines methods, showing that **synthetic captions, such as BLIP might generate, are better prompts that human-designed captions**.

**Ablations**   Table 5 shows the results of two ablations. First, we report BLIP-CLIP on CelebA, which OpenCLIP does poorly on (as shown earlier in Table 1). We see that BLIP-CLIP leads to high accuracy on this dataset. Next, we also ablate for using multiple templates on Waterbirds, which is a common practice done by CLIP practitioners. Specifically, we report the accuracy averaged over the following four preamble templates: *"This is a picture of the bird called a"*, *"This is an image of the bird called a"*, *"This is an picture of the bird known as a"*, and *"This is an image of the bird known as a"*. Table 5 shows that BLIP-CLIP retains high accuracy and is robust to the choice of template. This is important not only in showing that BLIP-CLIP works with template averaging, but also that

| | CelebA | | | Waterbirds (Multiple Templates) | | |
|---|---|---|---|---|---|---|
| | Worst-Group | Average | Gap | Worst-Group | Average | Gap |
| Baseline Zero-Shot | 15.6 | 89.0 | 73.5 | 39.3 | 77.2 | 38.0 |
| BLIP-CLIP Image-to-Caption $k$-NN | **76.2** | 79.9 | 3.7 | **71.2** | 79.3 | 8.1 |

Table 5: Ablation results. For the CelebA ablation, the baseline zero-shot results are from OpenCLIP. We note that the original CLIP models perform well on CelebA, as shown in Table 1. For the second ablation, the baseline zero-shot results are for the CLIP ResNet-50 model.

CLIP's sensitivity to text prompts that we identify in Section 4 is **due to keyword prompts being OOD and not simply from other arbitrary choices on text such as using a different template.**

## 6 DISCUSSION AND CONCLUSION

Our work is the first to investigate the unexplained differences in spurious correlation behavior between CLIP and unimodal vision models. In doing so, we uncover the key finding that the choice of text prompts matters greatly for zero-shot robustness, with CLIP's performance suffering when OOD class prompts are used. This is especially harmful in spurious correlation datasets, where the OOD prompts can *reinforce* spuriousness. We note that our results and our proposed approach are not restricted to CLIP or even CLIP-like models: they can be extended to multimodal generative models more generally, where text is one of the modes of information. As the ImageNet-1K experiment shows, our work also extends to broader distribution shift tasks beyond spurious correlations, so long as the distribution shift arises from OOD text prompts.

**Future Work**    (1) Testing BLIP-CLIP on non-spurious correlation datasets, to understand if BLIP-generated captions are universally useful in improving accuracy even when spurious correlations do not exist. (2) BLIP-CLIP is not fully automated as there is still some manual input in the form of choosing a reasonable preamble prompt to query BLIP for completion. An automated system for choosing the preamble prompt will be useful.

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

## A EXPERIMENTAL DETAILS

**Datasets**   For both datasets, we follow the standard test/train splits. Waterbirds contains 4796 training examples and 5794 test examples. CelebA contains 19962 test examples; we do not use the training set in CelebA throughout this paper.

**CLIP**   For all experiments, we use the pretrained CLIP implementation from `https://github.com/openai/CLIP` out of the box. Unlike Zhang & Ré (2022) and Yang et al. (2023), we experiment with the ResNet-50 and ViT-L/14 image encoder architectures.

**BLIP and Llama-2**   We use the implementation of BLIP from Hugging Face (`https://huggingface.co/Salesforce/blip-image-captioning-large`) and the official implementation of Llama-2 from `https://github.com/facebookresearch/llama`.

**Class Prompts**   We follow the same prompt templates as Zhang & Ré (2022) for all experiments in Section 4 except for the CelebA in Table 3 where we intentionally make changes to the class prompts. For Waterbirds, we use the preamble *"This is the image of a [class_name]."* For CelebA, we use the preamble *"A photo of a celebrity with {blond, no blond} hair"*.

**Section 4.4**   We plot $\log \tilde{q}(\mathbf{t}_{[2]}|\mathbf{t}_{[1]})$, where $\mathbf{t}_{[1]}$ is the preamble template *"This is a picture of"* that we have used for Waterbirds and $\mathbf{t}_{[2]}$ is the completion of interest. We plot 72 choices of $\mathbf{t}_{[2]}$ — the words *"waterbird"* and *"landbird"* themselves, as well as 70 fine-grained bird names. [2] Figure 2 shows that the word *"landbird"* has one of the lowest likelihoods under Llama-2, lower than almost all 70 specific bird names. The word *"waterbird"* has higher likelihood but is still less probable than half of the specific bird names.

---

[2]These 70 bird names are selected by taking the last word of all 200 species of birds in the dataset and removing duplicates, i.e. different species of birds in the same family will be mapped to a single point. This ensures a fair comparison to *"waterbird"* and *"landbird"*, which are themselves one-word prompts.

## B  FURTHER RESULTS

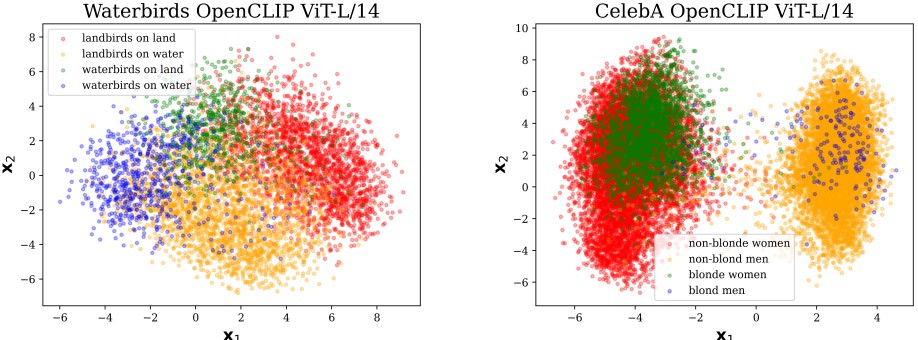

Figure 3: 2D projections of OpenCLIP image embeddings for the Waterbirds and CelebA test sets, colored by group.

