# OpenReview forum: "Unclipping CLIP's Wings: Avoiding Robustness Pitfalls in Multimodal Image Classification"
_ICLR.cc/2025/Conference — Submitted to ICLR 2025_

### Official Review · Reviewer_Kx98 · 2024-10-17

**Soundness:** 3
**Presentation:** 3
**Contribution:** 3
**Rating:** 8
**Confidence:** 3

**Summary:**

The paper studies the robustness of CLIP models on datasets with spurious correlations (Waterbirds, CelebA).

The authors find that the zero-shot classification performance is highly dependent on the choice of class prompts

They then present a method to improve this, using class prompts generated by an image-to-text model. They show that this works better because the prompts are "ID" w.r.t. CLIP's pretraining data.

**Strengths:**

- Clear writing, thorough presentation of the problem, methods, and results. The paper covers particularly well the context of the problem (spurious correlations, shortcut learning) and the current state of knowledge about how CLIP models behave and are currently being used.

- Scientific evaluation of a well known problem with CLIP models

- Simple mitigation method that appears effective, and also seem practically relevant to many uses since it allows using CLIP as a classifier without manually creating text prompts/class descriptions.

- Evaluation of several CLIP variants (OpenCLIP, MetaCLIP)

- Preliminary demonstration of usefulness on data without spurious correlations (IN-1k)

**Weaknesses:**

I do not see major flaws in this paper. It has interesting findings that (to my knowledge) are novel, and a simple method that seems practically useful.

The findings are not earth-shattering but I think they will be of interest to the community working on spurious correlations and on the robustness of CLIP models.

**Questions:**

Presentation tips:

- "WGA can be arbitrarily improved": In the abstract, this sentence should be rephrased. The current sentence means that it could be improved up to 100%, which is not the case I believe.

- Tables 3/4/5 do not fit the width of the page. The tables might also be nicer with the booktabs package (with \toprule, \midrule, and \bottomrule) and possibly fewer vertical lines (?).

---

> ### Author Response · Authors · 2024-12-03
> **Response to Reviewer Kx98**
>
> We thank you greatly for the positive review and kind support of our work. We invite you to read through our comments to other reviewers. We have taken note of the comments you have made regarding presentation and will clean them up for a future version of the manuscript.

---

### Official Review · Reviewer_T6Nm · 2024-11-03

**Soundness:** 2
**Presentation:** 3
**Contribution:** 2
**Rating:** 3
**Confidence:** 4

**Summary:**

The paper investigates the performance of CLIP models on datasets containing spurious correlations. First, the paper shows that multiple CLIP models have poor zero-shot performance when predicting the target class or the spurious attribute in the considered datasets (Waterbirds, CelebA). A cause of this is that the class/attribute labels are unusual, out-of-distribution of the training captions. The proposed solution is to not rely on the target name, but instead use an image captioning model (BLIP) to get a caption for each image and then embed it with the CLIP text module. A testing image will be encoded either by: a) the CLIP image encoder or b) captioning the image with BLIP and then using the CLIP text encoder. This embedding will be used in a k-NN fashion together with the previous support set.

**Strengths:**

* S1. Investigating the biases of CLIP models is a good direction.

* S2. Investigating the frequency of class labels in LAION dataset and looking at the marginal log-probabilities given by MetaCLIP models is interesting and suggests that the class names were not frequent in the training datasets.

**Weaknesses:**

* W1. The message of the paper is not that novel. The community knows that large models are very sensitive to prompts, it is not new or surprising that CLIP performance varies for class templates.

* W2. Waterbirds and CelebA are small datasets and having most of the analysis done on them is a big limitation. Fundamental research questions could still be studied on these datasets, but investigating the robustness prompts / templates and proposing complex solutions involving multiple large models to solve these datasets is hard to justify. It’s hard to say if these investigations can lead to insights and benefits for more realistic settings.

* W3. There are simple baselines missing. a) Linear evaluation on CLIP image features b) k-nn directly on CLIP image features c)  same for BLIP image features. These are necessary to see if the captioning is important for robustness or if simply using the good features of CLIP / BLIP is enough.

* W4. It is common to use image captions and LLMs to solve different tasks (see Blip2 or LLaVA). Using image captions from large models like BLIP to help in CLIP classification via k-NN seems like a step behind such approaches, without giving any additional insights.

* W5. Papers like [A] also investigate the non-robustness of CLIP class templates and suggest instead to use detailed descriptions generated by LLM as templates. It will be relevant to discuss the similarities to the proposed solution.

[A] Menon, Sachit, and Carl Vondrick. "Visual classification via description from large language models."

**Questions:**

How many captions are used for each image? Were there other models than BLIP testesd for captioning?

---

> ### Author Response · Authors · 2024-12-03
> **Response to Reviewer T6Nm**
>
> We thank you for taking the time to read and leave a thoughtful review of our work. We will respond to the Weaknesses and Questions sections below:
>
> **W1 and W2:** We agree that the insight that “CLIP is sensitive to the choice of prompts” itself is not novel, as Reviewer 1 has also pointed out. However, we believe our work is the first to highlight the implication of this on studying spurious correlations in CLIP. Here, we want to address why we believe making this connection is important.
>
> Waterbirds and CelebA are both benchmark datasets used by the spurious correlation community. They feature prominently in the unimodal vision setting, used in every single one of the works we referenced in our Related Work section.
>
> In contrast, spurious correlations in CLIP models is a relatively new area of research. The few existing works in the literature, such as [1] and [2], also make use of both Waterbirds and CelebA. We believe the value of our work is showing precisely that these datasets are poor benchmarks for multimodal research — when minor changes to text prompts can cause as much as 30% changes to WGA, it should be clear that the efficacy of existing methods should be scrutinized.
>
> Considering how heavily Waterbirds and CelebA are used by the community, we believe it is important to stress this point. A dataset whose baseline performance swings by as much as 30% cannot be a valid benchmark. We emphasize that the implications of such research extend beyond performance on these datasets. For example, one key question as we move from the unimodal setting into the multimodal setting is whether large, pretrained vision models suffer just as much from spurious correlations, and consequently, whether scaling does help to mitigate these effects [3]. Research directions like these will be hamstrung if we are using the wrong benchmark datasets.
>
> In this light, we indeed fully agree with the point you are making in W2, which is that Waterbirds and CelebA are small, non-natural datasets that have become poor benchmarks as we graduate from the unimodal vision setting to large, pretrained multimodal models like CLIP. However, this particular understanding is not at all discussed in the community, and we believe our work is an important step in the correct direction.
>
> **W3:** Baseline (a), linear evaluation, can be found in [1]. We fully agree with the need to add Baseline (b) and will add this to a future version of the manuscript.
>
> **W4:** We note that the kind of large pretrained models noted in W4 solve a different problem from the one we examine in our work. While we agree that it is possible to combine these models for more sophisticated capabilities, which BLIP-2 and LLaVA do, the solution presented in Section 5 of our work is specifically designed to tackle the problem of zero-shot image classification in CLIP. Even if the problem itself is a “simple” one, we note that spurious correlations in CLIP models (and in fact, spurious correlations in general) still remain a challenging and open problem even today.
>
> **W5:** We agree with your characterization of [A] in W5. However, we note that the type of distribution shifts in [A] relate to the transfer learning from one dataset to another. Spurious correlations, which we consider in our work, is a very different type of OOD phenomenon arising from inductive biases to learning specific (but erroneous) features that are correlated with the salient feature. We find it gratifying that the authors of [A] came to a similar conclusion on a different kind of OOD problem.
>
> **Q1:** We have used the entire dataset for captioning thus far. However, we also present an ablation below for the a smaller dataset size:
>
> |                | **Worst-Group Acc.** | **Average Acc.** |
> |----------------|----------------------|------------------|
> | $$\alpha=1.0$$ | 70.7                 | 80.4             |
> | $$\alpha=0.9$$ | 68.8                 | 78.8             |
> | $$\alpha=0.5$$ | 64.3                 | 75.2             |
>
> ---
>
> [1] Michael Zhang and Christopher Ré. Contrastive adapters for foundation model group robustness. Advances in Neural Information Processing Systems, 35:21682–21697, 2022.
>
> [2] Yu Yang, Besmira Nushi, Hamid Palangi, and Baharan Mirzasoleiman. Mitigating spurious correlations in multi-modal models during fine-tuning. arXiv preprint arXiv:2304.03916, 2023.
>
> [3] Wang, Qizhou, et al. "A Sober Look at the Robustness of CLIPs to Spurious Features." The Thirty-eighth Annual Conference on Neural Information Processing Systems.

---

### Official Review · Reviewer_iubQ · 2024-11-04

**Soundness:** 2
**Presentation:** 3
**Contribution:** 2
**Rating:** 3
**Confidence:** 3

**Summary:**

The paper investigates the robustness of CLIP zero-shot classification to spurious correlations and identifies out-of-distribution class prompts as possible cause. It proposes a mitigation by generating proxy class prompts using an image-to-text model on the downstream training data and performing k-nearest neighbours on the resulting embeddings.

**Strengths:**

The sensitivity of CLIP zero-shot classification to the choice of class prompts is a known problem. Improving the performance by choosing more suitable class prompts is computationally very efficient and thus an interesting alternative to methods based on fine-tuning. The lacking coverage of used class prompts in the original training data of the CLIP model seems to be a reasonable explanation for bad performance on downstream classification tasks and is to some extent supported by the experimental evidence. Further, the proposed method is computationally efficient and does not require spurious feature annotations.

**Weaknesses:**

- The proposed method improves over zero-shot classification but performs significantly worse compared to other methods using training data of the downstream task.
- The initial experiments show that the CLIP models perform much worse on Waterbirds compared to CelebA leading to the conclusion that lacking coverage of the prompts “waterbird” and “landbird” in the pertaining data are the cause for this discrepancy. However, for Waterbirds, the spurious labels “water” and “land” are contained in these class prompts which might cause a larger CLIP similarity between the corresponding text embeddings and the image embeddings containing “water”/“land” backgrounds. This is not the case for CelebA (class attributes: blond/not blond, spurious attributes: male/female). This simple explanation would cause a similar discrepancy but is not considered in the paper. Thus, the experiments on Waterbirds do not seem suitable to properly support the claims regarding lacking coverage.

Minor: Table 2, 3, 4, and 5 are too wide and go over the margin. The plots in Figure 1 are too small, axis labels and legend are not readable. Section 5 mentions the use of Llama-2 in Fig. 2 which contradicts its caption and description in section 4.

**Questions:**

As I understand Section 5, all samples from the downstream training set are used for the knn algorithm. How strong is the effect on the results if a smaller subset is used instead?

For the experiments on ImageNet-1K, the training set is used for evaluation. Was the training set further split into train and test data or were the evaluated images also used to create the text-embeddings of the BLIP captions?

---

> ### Author Response · Authors · 2024-12-03
> **Response to Reviewer iubQ**
>
> We thank you for taking the time to read and leave a thoughtful review of our work. We will respond to the Weaknesses and Questions sections below:
>
> **W1:** We agree that experimental results for Waterbirds are not the strongest. However, we stress that the main contribution of the paper should be Section 4. Waterbirds and CelebA are both benchmark datasets used by the spurious correlation community. They feature prominently in the unimodal vision setting, used in every single one of the works we referenced in our Related Work section.
>
> In contrast, spurious correlations in CLIP models is a relatively new area of research. The few existing works in the literature, such as [1] and [2], also make use of both Waterbirds and CelebA. We believe the value of our work is showing precisely that these datasets are poor benchmarks for multimodal research — when minor changes to text prompts can cause as much as 30% changes to WGA, it should be clear that the efficacy of existing methods should be scrutinized.
>
> Considering how heavily Waterbirds and CelebA are used by the community, we believe it is important to stress this point. A dataset whose baseline performance swings by as much as 30% cannot be a valid benchmark. We emphasize that the implications of such research extend beyond performance on these datasets. For example, one key question as we move from the unimodal setting into the multimodal setting is whether large, pretrained vision models suffer just as much from spurious correlations, and consequently, whether scaling does help to mitigate these effects [3]. Research directions like these will be hamstrung if we are using the wrong benchmark datasets.
>
> In this light, Section 5, for us, represents one possible alternative that bypasses prompt sensitivity. We don’t intend for it to be state-of-the-art. Instead, it is a proof-of-concept that methods that are resilient to the choice of prompt can work too. Future research into spurious correlations in CLIP and other large pretrained models can take entirely different approaches besides prompt engineering. For example, future work can simply promote other large natural image datasets with valid spurious correlations that are invariant to the choice of prompts.
>
> **W2:** We respectfully disagree that the explanation presented here can adequately address the discrepancy between Waterbirds and CelebA. First, we note that we have already addressed this hypothesis in Table 2 of the paper, where we tried to predict Waterbirds’ spurious attribute itself. Table 2 shows that CLIP fails even at predicting the *background* itself, which does not lend credence to the idea that having “water” or “land” in the prompts cause the model to leverage background features for prediction. As for CelebA, Table 3 also shows that the seemingly good performance of CLIP on CelebA is a myth, with arbitrary changes to prompt reducing WGA by as much as 30%. As such, we disagree with the reviewer’s alternative explanation here for the discrepancy between Waterbirds and CelebA.
>
> **W3:** We will fix these formatting issues in a future version of the manuscript. The reference to Llama-2 is an erroneous leftover from a previous version of the paper that we removed.
>
> **Q1:** We can show the ablation over the size of the downstream training dataset used for kNN classification below. We show results for Waterbirfs below:
>
> |                | **Worst-Group Acc.** | **Average Acc.** |
> |----------------|----------------------|------------------|
> | $$\alpha=1.0$$ | 70.7                 | 80.4             |
> | $$\alpha=0.9$$ | 68.8                 | 78.8             |
> | $$\alpha=0.5$$ | 64.3                 | 75.2             |
>
> **Q2:** We used the validation set as the “training data”, i.e. to create the embeddings.
>
> ---
>
> [1] Michael Zhang and Christopher Ré. Contrastive adapters for foundation model group robustness. Advances in Neural Information Processing Systems, 35:21682–21697, 2022.
>
> [2] Yu Yang, Besmira Nushi, Hamid Palangi, and Baharan Mirzasoleiman. Mitigating spurious correlations in multi-modal models during fine-tuning. arXiv preprint arXiv:2304.03916, 2023.
>
> [3] Wang, Qizhou, et al. "A Sober Look at the Robustness of CLIPs to Spurious Features." The Thirty-eighth Annual Conference on Neural Information Processing Systems.

---

### Official Review · Reviewer_Sz33 · 2024-11-04

**Soundness:** 2
**Presentation:** 1
**Contribution:** 1
**Rating:** 3
**Confidence:** 5

**Summary:**

This paper demonstrates through experiments that factors such as dataset bias can introduce spurious correlations in pre-trained models. To mitigate these spurious correlations, the paper proposes using captions generated by BLIP as class prompts. Experimental results show that using BLIP-generated class prompts can enhance CLIP’s robustness to spurious correlations without requiring fine-tuning.

**Strengths:**

This paper demonstrates, through performance comparisons, feature visualizations, and dataset statistics, that the choice of prompts has a significant impact on zero-shot performance. The authors also propose a BLIP-CLIP architecture that enhances CLIP’s zero-shot capability without requiring fine-tuning.

**Weaknesses:**

1. First, the author should adjust the layout of the tables in the paper. Some tables exceed the page width. (Tables 2, 3, 4, and 5).

2. This paper demonstrates in multiple ways that CLIP is sensitive to prompts. Previous work, such as MaPLE [1], had already presented similar insights.


3. Although the method in this paper requires no training, it involves comparing the similarity between each test sample and the captions generated for all training samples during the inference stage. On a large-scale dataset, wouldn’t this approach introduce a substantial additional time cost? It would be helpful if the authors could provide a comparison of the time cost between this method and direct inference with CLIP. Have you considered optimizing the computational burden on large datasets?

4. The experiments in this paper seem to focus more on the classification of spurious correlations. For the experiments on ImageNet, only 13 different cat categories were selected from the 1,000 classes to classify Pantherinae and Felinae. Since the method proposed in this paper does not require retraining, why didn’t conduct a performance evaluation directly on the entire ImageNet dataset? FD-Align [2] also pointed out that fine-tuning directly on ImageNet can impact model generalization due to spurious correlations, which is consistent with the distribution shift caused by OOD text as proposed in this paper.


[1] MaPLe: Multi-modal Prompt Learning

[2] FD-Align: Feature Discrimination Alignment for Fine-tuning Pre-Trained Models in Few-Shot Learning

**Questions:**

1. In the 4th row of Table 1, OpenCLIP shows high average performance and spurious prediction on CelebA, yet the Worst Group Accuracy is very low. If this is due to dataset bias, how can the high performance in spurious prediction be explained?

---

> ### Author Response · Authors · 2024-12-03
> **Response to Reviewer Sz33**
>
> We thank you for taking the time to read and leave a thoughtful review of our work. We will respond to the Weaknesses and Questions sections below:
>
> **W1:** We will fix this in a future version of the manuscript.
>
> **W2:** We agree that the insight that CLIP is sensitive prompts itself is not novel. However, we believe our work is the first to highlight the implication of this on studying spurious correlations in CLIP. Here, we want to address why we believe making this connection is important.
>
> Waterbirds and CelebA are both benchmark datasets used by the spurious correlation community. They feature prominently in the unimodal vision setting, used in every single one of the works we referenced in our Related Work section.
>
> In contrast, spurious correlations in CLIP models is a relatively new area of research. The few existing works in the literature, such as [1] and [2], also make use of both Waterbirds and CelebA. We believe the value of our work is showing precisely that these datasets are poor benchmarks for multimodal research — when minor changes to text prompts can cause as much as 30% changes to WGA, it should be clear that the efficacy of existing methods should be scrutinized.
>
> Considering how heavily Waterbirds and CelebA are used by the community, we believe it is important to stress this point. A dataset whose baseline performance swings by as much as 30% cannot be a valid benchmark. We emphasize that the implications of such research extend beyond performance on these datasets. For example, one key question as we move from the unimodal setting into the multimodal setting is whether large, pretrained vision models suffer just as much from spurious correlations, and consequently, whether scaling does help to mitigate these effects [3]. Research directions like these will be hamstrung if we are using the wrong benchmark datasets.
>
> **W3:** We will add this ablation in a future version of the work.
>
> **W4:** We chose a subset of ImageNet specifically to highlight the scenario where prompt OOD is an issue on a natural image dataset. We accept the point made here and will add further results on the entire ImageNet dataset in a future version of the work.
>
> **Q1:** It is unclear why the ViT architecture for OpenCLIP perform so poorly on CelebA. However, we note in Table 3 that regardless of the pretraining dataset, small changes in prompts can still result in high sensitivity to WGA. This is true on both CLIP and OpenCLIP, albeit in different directions. However, our underlying conclusion is still valid, which is that prompt sensitivity can arbitrarily mitigate or worsen the effects of spurious correlations, and more poignantly, that the results of existing research should be scrutinized.
>
> ---
>
> [1] Michael Zhang and Christopher Ré. Contrastive adapters for foundation model group robustness. Advances in Neural Information Processing Systems, 35:21682–21697, 2022.
>
> [2] Yu Yang, Besmira Nushi, Hamid Palangi, and Baharan Mirzasoleiman. Mitigating spurious correlations in multi-modal models during fine-tuning. arXiv preprint arXiv:2304.03916, 2023.
>
> [3] Wang, Qizhou, et al. "A Sober Look at the Robustness of CLIPs to Spurious Features." The Thirty-eighth Annual Conference on Neural Information Processing Systems.

---

### Author Response · Authors · 2024-12-03
**General Response**

We thank all reviewers for their time and effort in reviewing our work and leaving thoughtful comments and suggestions. As noted by all reviewers, there are multiple strengths of the paper:
- Investigating biases of large pretrained models, such as CLIP, is important, and our work is well-motivated in studying spurious correlations learnt by such models.
- Our work solidly demonstrates, through a combination of methods (feature visualization, analyzing data log-probabilities, and comparing accuracies on different prompts), that CLIP is not only sensitive to class prompts, but also that this sensitivity greatly affects spurious correlations, creating swings in accuracy by as much as 30%.
- We propose a computationally efficient alternative to fine-tuning the downstream dataset. Our method is not sensitive to the choice of prompt and reflects a more sober and accurate look at these spurious correlation datasets.

We believe that our work is a timely and important contribution to spurious correlation research, especially so in the intersection of spurious correlation and large pretrained models, which has not been studied extensively.

Beyond the method that we have proposed, we believe that the most important contribution of our work is to **acknowledge the flaws in existing benchmark datasets such as Waterbirds and CelebA**. These datasets have been widely studied in past literature. We show that they are increasingly poor benchmarks as we move to larger models like CLIP, and our work is an important step towards more robust analyses in spurious correlation research.

We respond to criticisms individually to each reviewer below.

---

### Meta-Review · Area_Chair_fCUi · 2024-12-08

**Metareview:**

The paper investigates robustness issues in CLIP models when applied to datasets with spurious correlations (e.g., Waterbirds, CelebA), focusing on sensitivity to prompt design. The authors propose using BLIP-generated prompts combined with a k-NN approach to improve performance without fine-tuning. While the paper raises relevant concerns about dataset biases and spurious correlations, significant weaknesses outweigh its contributions.

Strengths:
* The paper provides a detailed investigation into the sensitivity of CLIP models to class prompts, showing how minor changes can significantly affect performance, particularly in the context of spurious correlations.
* The proposed method using BLIP-generated class prompts to improve robustness without fine-tuning is seen as novel and practically useful by some reviewers.
* The work contributes to understanding the limitations of benchmark datasets like Waterbirds and CelebA when applied to multimodal models like CLIP.

Weaknesses:
* The novelty of the core findings regarding prompt sensitivity in CLIP is questioned, as this sensitivity has been previously noted in the literature.
* The reliance on small, potentially unrepresentative datasets like Waterbirds and CelebA for the majority of the analysis is seen as a significant limitation.
* There's a lack of comparison with simple baselines, which could have provided clearer insights into the effectiveness of the proposed method.
* Some reviewers noted issues with the presentation, like tables extending beyond page margins and the need for better formatting.

Given the reviews, the paper's contribution to the field, while interesting, does not introduce sufficiently novel insights or methodologies beyond what is already known about CLIP's sensitivity to prompts. The proposed method, although practical, lacks comprehensive benchmarking against simpler alternatives, and the choice of datasets limits the generalization of findings.

**Additional Comments On Reviewer Discussion:**

* Prompt Sensitivity (R1, R3, R4): Prompt sensitivity in CLIP is well-known; limited novelty in findings. Authors argue their contribution lies in connecting prompt sensitivity to spurious correlation benchmarks.
* Dataset Limitations (R1, R3): Heavy reliance on small datasets (Waterbirds, CelebA); concerns about generalizability. Authors agree but stress these datasets’ widespread use in spurious correlation research.
* Baseline Comparisons (R3): Missing simpler baselines like linear evaluation or k-NN on CLIP features. Authors promise to include these in future work.
* Computational Overhead (R1, R2): k-NN approach criticized for inefficiency on large datasets. Authors acknowledge and plan further ablations.
* Presentation Issues (All): Poor formatting of tables and figures; unclear phrasing in abstract. Authors promise to fix formatting and rephrase unclear statements.

---

### Decision · Program_Chairs · 2025-01-22

Reject